# Chronic Kidney Disease, Urinary Tract Infections and Antibiotic Nephrotoxicity: Are There Any Relationships?

**DOI:** 10.3390/medicina59010049

**Published:** 2022-12-27

**Authors:** Ioana Dicu-Andreescu, Mircea Niculae Penescu, Cristina Căpușă, Constantin Verzan

**Affiliations:** 1Clinical Department No 3, Internal Medicine and Nephrology, “Carol Davila” University of Medicine and Pharmacy, 050474 Bucharest, Romania; 2“Dr. Carol Davila” Teaching Hospital of Nephrology, Nephrology Department, Cal. Grivitei No 4, Sector 1, 010731 Bucharest, Romania

**Keywords:** chronic kidney disease, urinary tract infections, asymptomatic bacteriuria, antibiotic nephrotoxicity

## Abstract

Chronic kidney disease (CKD) has been a constant burden worldwide, with a prevalence of more than 10% of the population and with mortality reaching 1.2 million deaths and 35.8 million disability-adjusted life years (DALYs) in 2017, as it is claimed by the Global Burden of Diseases. Moreover, an increase in its prevalence is expected in the next years due to a rise in the number of people suffering from obesity, diabetes mellitus and hypertension. On the other hand, with cardiovascular morbidity and mortality showing a downward trend, maybe it is time to focus on CKD, to minimize the preventable risk factors involved in its progression toward end-stage kidney disease (ESKD) and to offer a better quality of life. Another major health burden is represented by infectious diseases, particularly urinary tract infections (UTIs), as it is considered that approximately 40–50% of women and 5% of men will have at least one episode during their lifetime. Additionally, CKD consists of a constellation of immunological and metabolical disturbances that lead to a greater risk of UTIs: increased apoptosis of lymphocytes, elevated levels of tumor necrosis factor α and interleukin 6, which lower the function of neutrophils and increased levels of uremic toxins like p-cresyl sulfate and indoxyl sulfate, which alter the adherence and migration of leukocytes to the sites of injury. Moreover, UTIs can lead to a more rapid decline of kidney function, especially in stages G3-G5 of CKD, with all the complications involved. Last, but not least, antibiotherapy is often complicated in this category of patients, as antibiotics can also negatively affect the kidneys. This review will try to focus on the particularities of the urinary microbiome, asymptomatic bacteriuria and UTIs and the subtle balance between the risks of them and the risks of antibiotherapy in the evolution of CKD.

## 1. Introduction

Chronic kidney disease (CKD) affects more than 10% of the population, with great social and economic impact [1]. It is defined by Kidney Disease Improving Global Outcomes (KDIGO) guidelines as the presence of structural abnormalities, most often represented by albuminuria, or functional, which means an estimated glomerular filtration rate (eGFR) under 60 mL/min/1.73 m^2^, any of them being present for at least three months [2]. At first view, CKD appears an easily diagnosed disease, but in the absence of active screening measures, most often patients are evaluated when they already have moderate, or even advanced CKD, and, thus, all the complications. Moreover, studies concluded that these patients have a particular risk of hospitalization compared with other frequent comorbidities [3]. A study conducted by Ishigami in 2019 reviewed the epidemiology of infectious diseases in CKD and found that the risk of hospitalization with infections was increased by 50% in stage G3 of CKD—eGFR under 45 mL/min/1.73 m^2^ and by 2–3 times in stages G4 and G5—eGFR under 30 mL/min/1.73 m^2^, and under 15 mL/min/1.73 m^2^, respectively, compared to stage G2 of CKD—eGFR between 60 mL/min/1.73 m^2^ and 90 mL/min/1.73 m^2^ [4].

Urinary tract infections (UTIs) affect almost 50% of women and 5% of men during their lifetime and are usually defined as more than 10^5^ colony-forming units/mL (CFU/mL) on culture with signs and symptoms of infection, like suprapubic pain, dysuria, or urgency [5]. However, more recent studies argue that even a number as low as 10^2^ CFU/mL can mean a UTI, especially in acutely symptomatic women [6,7]. Two other possible findings should not be overlooked: asymptomatic bacteriuria and pyuria. The first one is defined as more than 10^5^ CFU/mL, but with no signs or symptoms of urinary infection [6]. In women, it should be repeated after two weeks, because between 10–60% of them do not have persistent bacteriuria [8]. The second refers to the presence of a minimum of 10 leucocytes/μL or a minimum of 50 leucocytes/high-power field(HPF) from unspun urine, either due to UTIs or to an inflammation of the kidney [3]. These findings are of utmost importance because they can reveal not only individual particularities but also, as it is argued by some studies, an increased risk for the progression of CKD [9]. However, the mechanisms are yet to be fully understood.

In addition, it was noted that proteinuria, which is a very frequent finding in CKD and also the most important predictor factor of end-stage kidney disease (ESKD) over a 10-year period, is a particular risk factor when it comes about infectious diseases [10]. Between proteinuria measured as albumin-creatinine ratio (ACR) and the risk of hospitalization was found a directly proportional relationship, as patients in stage G2 of CKD, but with an ACR above 300 mg/g had the same infectious risk, if not even increased as patients in stages G3b or G4, but with no albuminuria [4]. Lastly, proteinuria was associated with a 10% greater risk of inferior respiratory tract infections and with a 30% increased risk of sepsis compared with patients with no proteinuria [4]. With these findings in mind, a question arises: what are the mechanisms by which chronic kidney disease increases the infectious risk? Are there any particularities or can we resume everything to uremic metabolic derangements?

## 2. Genetic and Immunological Aspects

CKD represents a risk factor for infections in general, and urinary tract ones in particular, through all the constellation of immunological, metabolical and inflammatory changes, on one hand, and because of frequent contact of the patients with medical services, on the other [11]. A fact to be kept in mind is that infectious diseases are the most common cause of hospitalizations and the second one of mortality in this category of patients [6].

It must be highlighted that genetics is important, as it is known that urinary tract infections, especially recurrent UTIs, often show familial aggregation [12]. The urinary tract is protected by the innate immune system and the susceptibility to UTIs is controlled by innate signaling pathways, which can be modified in some individuals by different genetic polymorphisms [12].

Of these, genes altering the expression of toll-like receptors (TLRs) are of the utmost importance, as TLRs have the key role in the protection of urothelium against urinary pathogens, by activating the innate immune response and a multitude of polymorphisms can enhance or decrease their function [13,14]. TLRs are trans-membrane protein molecules with two domains, expressed in a wide range of antigen-presenting cells, endothelial and epithelial cells [13]. After the pathogen manages to enter the urinary tract, TLRs recognize pathogen-associated molecular patterns (PAMPs) and then lead to a cascade of inflammatory responses in order to eliminate the intruder [13]. The result is an enhanced release of tumor necrosis factor alfa (TNFα), interleukin 1β (IL-1β) and interleukin 12 (IL 12), all with pro-inflammatory activity [15]. However, this effect is not always protective for the host, especially in case of systemic infections with Gram negative bacteria, as the lipopolysaccharide is recognized by TLR4 on both leukocytes and renal cells, which can favor the occurrence of acute kidney injury (AKI) via release of TNFα and IL1 with all the complications involved [15].

In humans, TLR2, TLR4 and TLR5 are the most involved in the protection against UTIs [12,13]. A detailed description of the role of TLRs and also the effect of different polymorphisms on the risk of UTIs is provided in Table 1.

Another notable fact is that polymorphisms involving vascular endothelial growth factor A (VEGFA) and transforming growth factor beta 1 (TGFβ1) were related to kidney scarring after UTIs and progressive kidney disease, especially associated with vesico-ureteral reflux [12,21]. VEGFA and TGFβ1 are in the center of tissue repair after different types of injury and polymorphisms like C-509 T allele in TGFβ1-promoter, or −460 CC in VEGF gene were more frequently associated with UTIs and kidney scarring [21]. However, it seems that they do not represent a risk factor for recurrent UTIs too, and, therefore, we can conclude that, from this point of view, if we can prevent UTIs, we can minimize the risk for progression even in the patients with these genetic variants [22]. Additionally, as a fact, in children with UTIs, the homozygosity of the C allele of the VEGF-A gene was associated with hypodysplastic renal parenchymal lesions [23]. Last, but not least, CXC chemokines and their receptors have an important role in providing an inflammatory local response to UTIs [24]. Polymorphisms in the interleukin 8-receptor (CXCR1), which is a crucial chemokine receptor involved in neutrophil recruitment, and, therefore, in an optimal function of the innate immune system, can be associated with UTIs, especially in children [25,26]. However, more research is needed, as evidence is conflicting [27]. Additionally, a low level of CXCR2 was found to be associated with a greater risk for UTIs, at least in premenopausal women [28].

Other important aspect which favors the multiplication of uropathogens is that in CKD there is a chronic pro-inflammatory status. The cellular immunity is defective and characterized by increased apoptosis of T lymphocytes CD4+ (LT CD4+) and elevated levels of TNFα and interleukin 6 (IL-6), which, counterintuitive, is not a protective mechanism [5,7,11,20]. Leukocytes are affected in multiple ways: firstly, TNFα and IL-6 lead to lower activity of neutrophils (although their absolute number remains the same), secondly, an elevated level of fibroblast growth factor 23 (FGF23) inhibits diapedesis, thirdly, uremic toxins like p-cresyl sulfate and indoxyl sulfate can alter the endothelial function, disturbing the adherence and migration of leukocytes to the injured area and, in the end, their phagocytic function is not maintained anymore because of alterations on the hexose-monophosphate shunt [5,7,10,20]. The increased levels of TNFα and IL-6 have even more implications: lower recruitment of macrophages, a higher threshold of reactive oxygen species (ROS) generation, which can minimize the clearance of bacteria and viruses and, in addition, two studies conducted by Ishigami et al. in 2019 and 2020 concluded that TNFα, IL-6 and C-reactive protein (CRP) are directly correlated with the risk of hospitalization for major infections, including UTIs [5,7].

Humoral immunity, defensins and uromodulin are also affected, with lower levels of IgM and secretory immunoglobulins (sIgA, sIgM), especially in uremic children, an altered release of them in the urine and a low capacity to protect against fimbriated bacteria being present [20,29]. In addition, the patient with CKD virtually always has other comorbidities, with their own impact on the immune system and on the protective capacity of the host against the massive number of infectious threats [4].

## 3. The Role and the Characteristics of the Urinary Microbiome, Pyuria, Asymptomatic Bacteriuria and UTIs

Although in the past it was believed that urine is sterile, nowadays, with the advancement of metagenomic analysis, we know that there are a lot of microorganisms that inhabit the urinary tract, with significant differences between healthy and sick people [30]. Additionally, the roles of microbiota, in general, were widely studied, and the conclusions should not be overlooked, as the alterations of microbial flora can lead to an altered immune system, inflammatory bowel disease, non-alcoholic fatty liver disease, type 2 diabetes mellitus, obesity and neurological damage [30,31].

Even though the urinary microbiome is not as extensively studied as the gut one, the most common members found are Corynebacterium, Lactobacillus, Ureaplasma, Jonquetella, Parvimonas, Proteiniphilum and Saccharofermentans, the last four being present especially after the age of seventy [31]. The only significant age difference between microorganisms was that Lactobacillus was more prevalent in pre-menopausal women, while Mobiluncus in post-menopause [32]. Additionally, there are some gender particularities, in the female urinary tract being more commonly seen Lactobacillus spp, while in men Corynebacterium and Streptococcus [30].

However, how can influence CKD the urinary microbiome? Or can it be the other way? An answer is attempted by a study conducted by Modena et al. in which healthy controls were compared with renal transplant recipients and patients who developed cellular rejection of the transplant, with interstitial fibrosis and tubular atrophy [33]. They found that Streptococcus, which is a dominant genus in healthy male controls was reduced in transplanted patients and even more in those with interstitial fibrosis and tubular atrophy [33]. Additionally, the diversity of the microbiome was found to be associated with eGFR, neurogenic dysfunction of the bladder, interstitial cystitis and urgency urinary incontinence [31,34]. As a particularity, in urgency urinary incontinence, the microbiome consisted of a much better-represented Gardnerella and a much lower level of Lactobacillus [35]. Probably the most plausible answer to the question is that urinary microbiome and CKD are interrelated and changes in one are reflected in the other. Further studies are needed to shed light on this aspect.

Another common finding that is worth looking at in CKD is pyuria; Cabaluna et al. rightly claimed from 1977 that it appears frequently in dialyzed patients [36]. Data is limited about how to correctly interpret it; nonetheless, in a more recent study, Kwon et al. identified pyuria in 30.5% of their cohort of patients with CKD (24.1% in non-dialyzed patients and 51.4% in hemodialyzed patients), the vast majority being sterile pyuria (over 70%) [37]. This is a reason why in CKD pyuria may not represent only a marker of UTI, but also one of parenchymal inflammation [3]. Additionally, in the general population, its prevalence is estimated at 7.2% (1.6% in men and 18.7% in women) [37].

There are a few conditions and characteristics that are correlated with the presence of pyuria: advanced age, female gender, diabetes mellitus, anemia, hypoalbuminemia, low eGFR, tubulointerstitial nephropathy, a pro-inflammatory status and, of course, the presence of a UTI [3]. Interestingly, glomerulopathies were not associated with pyuria [3]. Another finding is that hematuria and proteinuria also appeared more frequently in the pyuria group [37]. On the other hand, it may not be only a neutral finding in CKD, as can appear at first view, but also a possible marker of progression to ESKD and increased mortality, especially in autosomal dominant polycystic kidney disease, although there are no enough solid proofs, from randomized controlled trials [8,9,20,36]. Of note, if a patient has sterile pyuria and signs and symptoms of UTI, especially in endemic regions, urogenital tuberculosis should be suspected [11].

One step further up the ladder that can lead to UTIs is asymptomatic bacteriuria (ASB). It is defined as the presence of more than 10^5^ CFU/mL in cultures, but with no signs and symptoms of infection [8]. Its prevalence increases with age and affects predominantly female gender—over the age of sixty-five more than 15% of females have ASB and this number increases to almost 50% in those from long-term care facilities [8]. In CKD, its prevalence was found to be 6.6% [37]. ASB is favored by genetic alterations that affect the innate immune system and a consistently found polymorphism is that of toll-like receptors 4 (TLR4) [10,12]. The 2019 updated guidelines from the Infectious Diseases Society of America argue that it should be treated with antibiotics only in pregnant women in the first trimester and in patients who will undergo urological interventions [8]. One thing is certain: even if it is unclear if ASB is involved in a faster decline of renal function, under certain conditions it can progress to UTIs and these last ones are associated with increased risk for renal and systemic complications [8,37].

As it was already shown, CKD represents a risk factor for infections in general, because of the multiple immunological changes detailed, and for UTIs in particular, due to the associated comorbidities, advanced age and urinary obstruction [3]. The most common pathogen involved in UTI in this category of patients is *E. coli* and the greatest risk appears to be in diabetic nephropathy, nephrotic syndrome, autosomal dominant polycystic disease, urolithiasis and in patients receiving immunosuppressive treatment [11]. However, UTIs in CKD pose another threat: urosepsis. Several risk factors have been identified: diabetes mellitus, length of hospitalization, Klebsiella spp infection and the presence of an indwelling urinary catheter [38]. Additionally, the rate of mortality was thoroughly evaluated at 1, 2, 6 and 12 months and the trend steadily increased from 8.6% in the first month to 17.6% in a year [39]. Factors that predicted mortality at 30 days were Charlson score and infection with a strain of *E. coli* with resistance to 3rd generation cephalosporins [39]. In addition, it should be kept in mind that not only this severe complication is to be avoided, but, ideally, even uncomplicated UTIs, as studies claim increased risk for renal events and an accelerated progression of CKD in patients in stages G3-G5 with UTI [3,6,9]. These aspects are also detailed in Table 2.

## 4. Antibiotic Nephrotoxicity

As medicine evolved, the first antibiotics appeared and it seemed that the fight with the abundance of microorganisms can be won. However, they developed complex defense mechanisms and nowadays we are confronted with a few bacteria that are resistant to every class of antibiotics and a huge global burden [40]. Moreover, not all antibiotics can be used in all patients, they have to be adapted or even excluded per primam in case of comorbidities.

What is of great importance in CKD is that antibiotics, especially when misappropriately administered, can lead to a great degree of nephrotoxicity, through different mechanisms: interstitial nephritis, direct tubular toxicity and acute tubular necrosis (the most common cause of acute kidney injury), intratubular crystal deposits, immune dysfunction and a decreased renal perfusion [41,42]. What makes things even more difficult is the fact that CKD patients are often in contact with medical care services or undergo invasive procedures, all of which expose them to multi-drug resistant microorganisms, and thus a greater risk for nephrotoxicity [43].

There is a fact that has to be highlighted: the most commonly used antibiotics for UTIs are fluoroquinolones and co-trimoxazole because they reach the target urinary concentrations, but in CKD they must be adjusted to match the eGFR [44]. The problem is that in clinical practice studies have shown that in up to 64% of cases are prescribed in much higher doses, with all the risks of adverse reactions and deterioration of kidney function involved (Table 3) [44,45].

Additionally, the vast majority of antibiotics disrupt the activity of the benefic microbiome, especially the gut one [35]. For example, after administration of fluoroquinolones, there was a decrease in Bifidobacterium, Alistipes, Firmicutes, and an increase in Bacteroides, Blautia, Eubacterium and Roseburia [35]. A particular fact that deserves further study is the importance of Oxalobacter formigenes, a gut bacteria that is involved in the oxalate metabolism, decreasing its absorption from the intestine, and thus having a potential protective role in urinary lithiasis [35]. This species is not the only one involved in decreasing the absorption of oxalate, but also Lactobacillus spp (especially acidophilus, plantarum and brevis), Streptococcus thermophilus and Bifidobacterium infantis [31]. However, what is not known is the impact of prolonged courses of antibiotics and if there can be a link between them and urolithiasis, at least in this category of patients.

A possible solution for minimizing the antibiotic impact on the microbiome is to co-administer probiotics, such as Lactobacillus rhamnosus GR-1 or Lactobacillus reuteri, some studies arguing that they can be beneficial even to replace the antibiotics in special categories of patients, with uncomplicated UTIs [35]. In addition, if antibiotherapy is mandatory, nitrofurantoin has the least impact on the microbiome, being able to decrease *Clostridioides* spp. and increase Faecalibacterium [35].

All the above described factors—genetic, immunological, and drug-induced—can interact and contribute to the occurrence of urinary tract infection, the worsening of kidney function, and furthermore to complications and even death (Figure 1).

## 5. Conclusions

It seems that, at least theoretically, we have a lot of data about the interplay between microbiome, urinary tract infections and chronic kidney disease. We know that in CKD the urinary microbiome is associated with eGFR, that in kidney transplant recipients Streptococcus spp is decreased and also that antibiotherapy is a double-edged sword in what concerns the microbiome. We also know that a great number of polymorphisms that involve TLRs, CXC receptors, VEGF and TGFβ1 increase the risk of urinary tract infections in both CKD patients and in otherwise healthy ones and UTIs lead to a faster decline in renal function. Pyuria and asymptomatic bacteriuria deserve a deeper look in the future, as they are frequent findings and it is plausible to mediate different pathways that can tip the scales in certain conditions toward a faster decrease in eGFR. Last, but not least, CKD is a particular condition with a lot of metabolical and immunological changes—uremic toxins like p-cresyl sulfate and indoxyl sulfate, pro-inflammatory molecules such as TNFα, IL6, IL12, alterations of both cellular and humoral immunity, findings that close this vicious cycle, CKD creating the premises of UTIs and UTIs increasing the risk of progression towards ESKD.

Another crucial fact is represented by the nephrotoxicity of antibiotics, especially in the situation of a patient with CKD and UTI, who is already exposed to multiple aggressions. Moreover, it was found that in up to 64% of cases, the most commonly used antibiotics for UTIs—fluoroquinolones and co-trimoxazole were prescribed in much higher doses, unadjusted to eGFR. Because of this, it is possible that a part of the accelerated decline in kidney function found after UTIs is due to the inappropriate dosage and use of antibiotics. Additionally, a fact that should not be overlooked is that the consequences on the long-term of the changes in microbiome after antibiotic use are not well-known, but may play a role in modulating future immune responses and predisposing the host to other infectious threats. To shed light on this aspect a prospective, randomized study is necessary.

In conclusion, despite the fact that it seems we know a lot, in clinical practice things often go amiss, with mortality from CKD reaching 1.2 million. Moreover, the mortality from cardiovascular disease, cancer and chronic obstructive pulmonary disease decreased, but not that from CKD, and, if we take note that the second cause of death in this category of patients is represented by infectious diseases, then maybe it is time to integrate all the knowledge and, in further studies, to try to elucidate the mechanisms that stand behind the steps that lead to this.

## Figures and Tables

**Figure 1 medicina-59-00049-f001:**
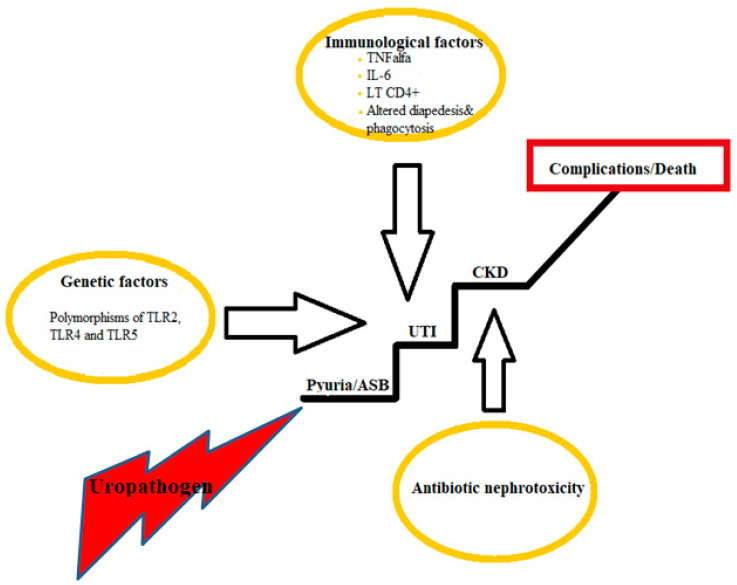
Factors potentially responsible for the worsening of kidney function in chronic kidney disease patients with urinary tract infections. ASB: asymptomatic bacteriuria; UTI: urinary tract infection; CKD: chronic kidney disease; TLR: toll-like receptor; TNF: tumor necrosis factor; IL: interleukin; LT: lymphocytes T.

**Table 1 medicina-59-00049-t001:** The distribution and roles of the toll-like receptors in humans.

Toll-like Receptors	Distribution	Brief Description and Characteristics	Role in UTIs	Source
**TLR1**	Cell membranes of epithelial cells and APCs	Forms heterodimers with TLR2 and TLR10;	Key role as heterodimer in UTIs caused by G(–) bacteria, especially UPEC; Polymorphism TLR1_G1805T confers protection against APN	[13,16]
**TLR2**	Cell membranes of epithelial cells and APCs	In homomeric form—no function; As heterodimers with TLR1, TLR6 and TLR10, recognizes molecular parts of LPS of non-Enterobacteriaceae	Polymorphisms TLR2Arg753Gln and TLR2 G2258A may represent a risk for rUTI, especially with G(+) bacteria	[11,13,14,17]
**TLR1-TLR2 heterodimers**	Cell membranes of epithelial cells and APCs		Recognizes triacylated lipoproteins of Mycoplasma, Ureaplasma and G(–) bacteria	[13]
**TLR2-TLR6 heterodimers**	Cell membranes of epithelial cells and APCs	Very similar to TLR1-TLR2 heterodimer (66% similarity, but different active ligand sites)	Recognizes the type of bacterial lipoproteins and peptidoglycan in G(+); useful in infections with *Staphylococcus* spp, *Streptococcus* spp, Mycoplasma, Ureaplasma	[13]
**TLR3**	Endolysosomes and in mature dendritic cells	Recognizes dsRNA in viruses	No role	[13]
**TLR4**	Cell membranes of the urothelium	Recognizes G(–) and intracellular bacteria, fungi and protozoa; The most important targets are LPS and type 1 and P fimbriae, which makes it particularly useful in UPEC	Protects urothelium against UTIs; Its level ↑ with the severity of the disease; Mutations and polymorphisms ↓ its activity, which allows the appearance of ASB; Polymorphism TLR4A299G may ↑ the risk for rUTI, especially with G(+) bacteria Conflicting evidence towards polymorphism TLR4_A896G: some argue that ↑ UTIs in children, and others that it is protective against rUTI;	[12,13,14,16,18]
**TLR5**	Cell membranes of epithelial cells and APCs	Protective role against UTIs	Recognizes flagellin, both from G(–) and G(+) bacteria; TLR5_C1174T polymorphism was associated with the intensity of the infectious disease and with rUTI (but not APN); Deletion leads to uncontrolled multiplication of bacteria in murine model	[13,14,16,19]
**TLR6**	Cell membranes of epithelial cells and APCs	Recognizes diacylated lipoproteins from bacteria and viruses	Useful in its heterodimeric form with TLR2	[13]
**TLR7, TLR8, TLR9**	Endolysosomes	Recognize genetic material from bacteria and viruses	No role in UTIs, but involved in controlling CMV infection after a kidney transplant	[13,20]
**TLR10**	Endolysosomes	Only in humans; Recognizes diacyl and triacyl lipoproteins	Role as a heterodimer in UTIs	[13]

TLR: toll-like receptor; UTIs: urinary tract infections; rUTI: recurrent UTI; G(–): Gram negative; G(+): Gram positive; APCs: antigen-presenting cells; UPEC: uropathogenic *E. coli*: APN: acute pyelonephritis; dsRNA: double-stranded RNA; LPS: lipopolysaccharides; ASB: asymptomatic bacteriuria; CMV: cytomegalovirus; ↓: decrease; ↑: increase.

**Table 2 medicina-59-00049-t002:** The role and the characteristics of urinary microbiome, pyuria, asymptomatic bacteriuria and UTIs.

Urinary Microbiome	Characteristics and Associations	References
Corynebacterium Streptococcus	Men > Women Reduced in post-transplant recipients and those with tubular atrophy or interstitial fibrosis	[30,33]
Lactobacillus	Pre-menopausal women *	[32]
Mobiluncus	Post-menopausal women *	[32]
Jonquetella Parvimonas Proteiniphilum Saccharofermentans	Over 70 years *	[31]
Ureaplasma	No specific characteristics	[31]
Gardnerella	↑ in urgency urinary incontinence	[35]

**Pyuria**	30.5% prevalence in CKD; 7.2%prevalence in general population. Correlated with: -advanced age; -female gender; -diabetes mellitus; -anemia; -hypoalbuminemia; -low eGFR; -tubulointerstitial nephropathy; -pro-inflammatory status; -presence of a UTI. Possible marker of progression in CKD, especially in ADPKD	[3,9,36,37]

**Asymptomatic bacteriuria**	6.6% prevalence in CKD; Associated with TLR4 polymorphisms; Unclear if it is involved in a faster decline of eGFR	[8,10,12,37]

**UTIs**	Greatest risk in: -diabetic nephropathy; nephrotic syndrome; -ADPKD; -urolithiasis; -immunosuppressive treatment; Can evolve to urosepsis, particularly in case of: -diabetes mellitus; -longer length of hospitalization; -Klebsiella spp infection; -presence of an indwelling urinary catheter. Can lead to accelerated progression of CKD in patients in stages G3-G5	[3,5,11,38]

* = more frequently, but not exclusively; > = more in the first category than in the other; ↑ = increase; CKD = chronic kidney disease; eGFR = estimated glomerular filtration rate; TLR4 = toll-like receptor 4; UTIs = urinary tract infections; ADPKD = autosomal dominant polycystic kidney disease.

**Table 3 medicina-59-00049-t003:** Interrelations of antimicrobial drugs and kidney.

Antibiotic	Renal Toxicity?	eGFR between 30–59 mL/min/1.73 m^2^	eGFR between 15–29 mL/min/1.73 m^2^	eGFR < 15 mL/min/1.73 m^2^	Source
**Nitrofurantoin**	No	Contraindicated by the manufacturer. New studies argue that it is safe for short-term	Contraindicated	Contraindicated	[46,47]
**Co-trimoxazole**	No	No adjustment	Reduce the dose to 50% of the usual dose	Reduce the dose to 25–50% of the usual dose or avoid it.	[48]
**Fosfomycin**	No	No adjustment	No adjustment	No adjustment	[49]
**Amoxicillin clavulanate**	Interstitial nephritis; Crystalluria	No adjustment	250–500 mg every 12 h	250–500 mg every 12 to 24 h	[50,51]
**Cefpodoxime**	Interstitial nephritis	No adjustment	The usual dose should be administered at 24 h	The usual dose should be administered at 48 h	[51,52]
**Cefdinir**	Interstitial nephritis	No adjustment	300 mg once daily	300 mg once daily	[51,53]
**Cefuroxime**	Interstitial nephritis	No adjustment	The usual dose should be administered at 24 h (in clinical practice there were no adverse reactions noted with unadjusted doses)	The usual dose should be administered at 48 h (in clinical practice there were no adverse reactions noted with unadjusted doses)	[51,54]
**Cefadroxil**	Interstitial nephritis	500 mg every 12 h	500 mg every 24 h	500 mg every 36 h	[51,55]
**Cephalexin**	Interstitial nephritis	No adjustment	250–500 mg every 8–12 h	250–500 mg every 12–24 h	[56]
**Ceftriaxone**	Interstitial nephritis	No adjustment	No adjustment	No adjustment (doses greater than 2 g have not been studied)	[57]
**Ciprofloxacin**	Interstitial nephritis, crystal-induced AKI (increased risk in association with ACEI)	250–500 mg every 12 h	500 mg every 24 h	500 mg every 24 h	[5,58,59,60,61]
**Levofloxacin**	Interstitial nephritis	No adjustment	250 every 48 h	250 every 48 h (risk of cardiac arrhythmias and sudden cardiac death in hemodialysis)	[51,62,63]
**Meropenem**	No Can cause AKI in combination with vancomycin	1 g every 12 h	500 mg every 12 h	500 mg every 24 h	[63]
**Vancomycin**	Oxidative tubular injury; AKI Interstitial nephritis	20–25 mg/kg loading dose; maintenance with 10–15 mg/kg at 24 h	20–25 mg/kg loading dose; maintenance with 10–15 mg/kg at 24 h	20–25 mg/kg loading dose; maintenance with 10–15 mg/kg at 48–72 h	[64,65,66]
**Piperacillin-tazobactam**	Debatable. A slightly increased risk for AKI, more elevated in combination with vancomycin	No adjustment	2.25 g every 6 h	2.25 g every 8 h	[67,68,69]

eGFR: estimated glomerular filtration rate; AKI: acute kidney injury; ACEI: angiotensin-converting enzyme inhibitors.

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
