# Peer review of "Chronic Kidney Disease, Urinary Tract Infections and Antibiotic Nephrotoxicity: Are There Any Relationships?"

_medicina, 2022, doi:10.3390/medicina59010049_

Round 1

Reviewer 1 Report

This review article discussed the possible relationship between chronic kidney disease, urinary tract infections and antibiotic nephrotoxicity. There are some issues in this manuscript as follows:

1.    Title: The word “disease” should be replaced with “diseases”.

2.    The novel points in this review article should be clarified because there are previous reviews that discussed the same  subject in more details; e.g. https://www.ncbi.nlm.nih.gov/pmc/articles/PMC8777485/; https://www.ncbi.nlm.nih.gov/pmc/articles/PMC7861116/

3.    Page 5 Lines 91-97: The sentence “CKD itself is a risk factor for infections in general, and urinary tract ones in particular, through all the constellation of immunological, metabolical and inflammatory changes, on one hand, and because of frequent contact of the patients with medical services, on the other[18]. A fact to be kept in mind is that infectious diseases are the most common cause of hospitalizations and the second one of mortality in this category of patients [5], mainly due to the alteration of both types of immunity – cellular and humoral, 96 and also of complement mediators[18].” has reference (18) that had been repeated. Please, add more recent references to this paragraph.

4.    Page 5 Lines 87 and 88: The role of the polymorphism involving vascular endothelial growth factor A (VEGFA) and transforming growth factor beta 1 (TGFβ1) in the progression of kidney disease should be explained in more details.

5.    Schematic figures for the immunological aspects of UTI in CKD and the role of the urinary microbiome, pyuria, asymptomatic bacteriuria and UTIs should be added.

6.    Page 6 Line 150: The word “”worth” should be replaced with “that worths)

7.    I think that the conclusion was insufficient. It should identify the possible clinical implications of the data obtained from the present review. Also, the conclusion should have no references.

8.    The manuscript should be thoroughly checked regarding the grammatical and typing errors.

Author Response

Good afternoon!

First of all, thank you for your time and good advice! It helped very much to better organize the manuscript.

  1. In the title we chose not to replace the word ,,disease`` with ,,diseases`` because we considered chronic kidney disease (CKD) as an entity, not different types of affections that can lead to CKD;
  2. Indeed, we were not very clear about the novel aspects of this article, but we hope to have fixed this, detailing more in conclusions about the often overlooked role of antibiotic use in UTIs in a CKD patient. Also, thank you very much for pointing out the two articles, but, even if our manuscript is not as detailed as them, it keeps the main points, focusses more on the genetic and immunological aspects  and also brings in front the nephrotoxic role of antibiotics and the frequent misuse of them in clinical practice. 
  3. We have changed this according to your suggestion, thank you!
  4. Also, we have provided more detailed explanations about the importance of polymorphisms of VEGFA and TGFbeta1 in recurrent UTIs, kidney scarring and, eventually, progression of CKD. 
  5. Indeed, that part of the manuscript lacked an easier form of presenting the information. We have added a table with all the main points and hope that it is much better know. Thank you!
  6. We have replaced the word ,,worth`` with ,,that worths``, according to your suggestion. 
  7. We eliminated the references from the conclusion and also detailed much more the pathways involved in the link between CKD, UTIs and antibiotic nephrotoxicity. Indeed, the conclusion in that form was insufficient.
  8. Also, we rechecked the manuscript for grammatical and typing errors and hope that it is better know. 

Thank you very much again for your effort and very good advice and hope that the changes made meet your demand.

Kind regards,

Ioana Dicu-Andreescu

Reviewer 2 Report

The manuscript ID medicina-2090654 entitled "Chronic kidney disease, urinary tract infections, and antibiotic nephrotoxicity: are there any relationships?" is a good review. Chronic kidney disease (CKD) has been a constant burden worldwide, with a prevalence of more than 10% of the population and with mortality reaching 1.2 million deaths and 35.8 million disability-adjusted life years (DALYs) in 2017, as it is claimed by the Global Burden of Diseases. Moreover, an increase in its prevalence is expected in the next years due to a rise in the number of people suffering from obesity, diabetes mellitus and hypertension. On the other hand, with cardiovascular morbidity and mortality showing a downward trend, maybe it is time to focus on CKD, to minimize the preventable risk factors involved in its progression toward end-stage kidney disease (ESKD) and to offer a better quality of life. Another major health burden is represented by infectious diseases, in particular urinary tract infections (UTIs), as it is considered that approximately 40-50% of women and 5% of men will have at least one episode during their lifetime. In patients with CKD, this type of infection is of particular importance because of three reasons: 1) CKD favors them due to a multitude of metabolic derangements, 2) UTIs can increase the risk of kidney function decline, especially in stages G3-G5 of CKD and last, but not least, 3) antibiotherapy is often managed with difficulty in this category of patients. This review will try to focus on the particularities of the urinary microbiome, asympto-matic bacteriuria and UTIs and the subtle balance between the risks of them and the risks of antibiotherapy in the evolution of CKD.

I appreciate the authors' effort in this review article. However, the following comments need to be addressed

1)      Abstract: There is a lack of connections of CKD between metabolic disorders and infectious diseases. The authors may rewrite it properly. Better to mention any pathways or cellular functions connecting these chapters.

2)      Background: What is KDIGO?;

3)      Unclear: "stage G2 of CKD, the risk of hospitalization with infections

4)      was increased by 50% in stage G3 and by 2-3 times in stages G4 and G5" define briefly G2 to G5 stages

5)      Genetic aspects: Here, authors need to elaborate key genes and pathways associated with CKD instead of TLRs, TLRs should be in immunological aspects 

6)      Differentiate or correlate TLRs, and UTIs with CKD clearly.

7)      Better to include any tabular form of information to "The role of the urinary microbiome, pyuria, asymptomatic bacteriuria, and UTIs"

8)      Figure 1 lacks contribution of pathogen info...to regulate or activate UTIs there must be the role of microbes and then it may modulate genetic and immunological functions or factors.

9)      Include more references up to 100 and give a mechanistic diagram to attract the audience to understand the authors' scientific hypothesis.

Author Response

Good afternoon!

First of all, thank you very much for your time and very valuable advice! It really helped to better organize the manuscript. 

  1. Indeed, there was a lack of connections between chronic kidney disease (CKD) and urinary tract infections (UTIs) in the abstract. We hope to have fixed this, as we followed your advice and detailed more about the mechanisms that make CKD to predispose patients to infectious threats. 
  2. KDIGO is the short form for Kidney Disease Improving Global Outcomes guidelines, but, as you pointed out, we did not provide in the text any explanation. Now we corrected this also. 
  3. The stage G2 was used for comparison of the risk of UTI in advanced stages of CKD. We also modified this to be more clear.
  4. As you suggested, we defined the stages of CKD, thank you!
  5. We detailed more the genes involved and also, to have better continuity, we did not separate anymore the genetic and the immunological aspects. 
  6. We provided better explanations for the role of toll-like receptors in UTIs and CKD in the text and also, the role of different polymorphisms in the table. Thank you for pointing this out!
  7. As you suggested, for a better understanding, we detailed the main aspects of the urinary microbiome, pyuria, asymptomatic bacteriuria and UTIs in a table. 
  8. Indeed, that aspect slipped our mind, we corrected the figure, thank you!
  9. In the end, we added more references, as we detailed more the aspects you pointed out. 

Thank you again for your effort in reading our manuscript and providing us very good suggestions! We hope that our changes meet your demand.

Kind regards,

Ioana Dicu-Andreescu